# SLIM-LLMs:
# Low-Rank Models of Linguistic Style*

## Abstract

Linguistic style encompasses a range of dimensions, including sensorial language as well as traditional stylistic features (represented using LIWC features). While these dimensions of linguistic style have been studied independently, relationships between the different dimensions, particularly between sensorial style and traditional stylistic features, remain understudied. This paper introduces a novel approach to model this interaction and tests it across a diverse set of texts. In particular, we propose using a Reduced-Rank Ridge Regression (R4) to model low-rank latent relationships between LIWC-based stylistic features and sensorial language features. We find that compared to the full LIWC feature set ($r = 74$), its low-dimensional latent representations ($r = 24$) effectively capture stylistic information relevant to sensorial language prediction. Based on our results, we propose Stylometrically Lean Interpretable Models (SLIM-LLMs) — dimensionality-reduced LLMs that model the non-linear relationships between these two major dimensions of style. We evaluate SLIM-LLMs on the ability to predict sensorial language (the actual sensorial words used) in five text genres: business reviews, novels, song lyrics, advertisements, and informative articles. Results show that SLIM-LLMs augmented with low-rank style features consistently outperform baseline models. These SLIM-LLMs approach the performance of full-scale language models while using significantly fewer parameters (up to 80% reduction).

## 1 Introduction

Salient and persistent structural patterns in language use constitute linguistic style. This includes traditional stylistic features like sentence length, language complexity, sentiment, and syntactic structure as well as patterns in the language used to describe sensory experiences — sensorial style. Sensorial style is a relatively new area of research and is informed by ideas from sensorial linguistics. Research into sensorial style holds significant potential for providing insights into human cognition.

Stylometrics, the study of linguistic style, has largely overlooked patterns in the use of sensorial language — words pertaining to sense and perception. While standard stylometric lexicons, such as LIWC, include some sensorial terms, these are generally distributed across different LIWC subcategories and their coverage of the sensorial language space is sparse. Moreover, there has been no focused investigation of the *relationship* between traditional stylometrics and sensorial language. Thus, we do not know for example if these two major dimensions of linguistic style are independent of each other or related to some degree. Our goal is to investigate this relationship.

Our motivation for studying this relationship stems from theories in cognitive science. The interaction between different dimensions of linguistic style can be modeled using cognitive frameworks similar to the 'mental lexicon' proposed by Levelt (1992), which posits a central repository of linguistic knowledge that mediates various aspects of language processing. We extend this idea to propose a *Central Language Processing Unit* (CLPU) that coordinates interactions between different representations of linguistic style.

---

*Our code and data are available at this anonymized Git repository:
https://github.com/iclranonsubmission/iclr_2024

This model of interactions within linguistic style also aligns with the grounded cognition theory (Barsalou, 2008), which suggests that linguistic processes are closely tied to the brain's perceptual, motor, and introspective systems. This theory implies that how we articulate sensory and bodily states influences our language use. However, previous research in this area has been limited to small-scale studies (Zwaan & Madden, 2005; Pulvermüller, 2013). In this context, our work aims to bridge this gap in stylometric research by computationally modeling the relationship between traditional style features and sensorial style across large and diverse text collections. We propose a novel approach to modeling this relationship, drawing from stylometrics, sensorial linguistics as well as cognitive sciences.

Our work makes the following contributions to the field:

- We model the interactions within traditional LIWC-style and sensorial style using Reduced-Rank Ridge Regression (R4). We use R4 to identify low-rank group structures within LIWC-style.
- We introduce Stylometrically Lean Interpretable Models (SLIM-LLMs), which provide a more interpretable lens to study the relationship between traditional linguistic style and sensorial style.
- We conduct large-scale analysis across diverse text genres, providing empirical support for theoretical claims about the interaction between different aspects of linguistic style.

## 2 RELATED WORKS

The study of sensorial style is a relatively new area of research. There are no directly comparable studies examining sensorial style and its relation to traditional styles. Instead, we review works from allied fields — stylometry and sensorial linguistics — that intersect with our work.

### 2.1 STYLOMETRY

Stylometry focuses on analyzing linguistic style use through various computational and statistical techniques. While much of stylometric research has centered on author attribution (Overdorf & Greenstadt, 2016), more recently stylometrics have been used to analyze emotional and psychological dimensions of language use.

One of the primary stylometric methods that focus on psycholinguistics is Linguistic Inquiry and Word Count (LIWC) (Pennebaker et al., 2007). LIWC measures various linguistic features, including emotional tone, cognitive processes, and personal concerns. It has been widely used for tasks ranging from author attribution to modeling psychological states such as depression (De Choudhury et al., 2013).

Similarly, ANEW (Bradley & Lang, 1999), provides a set of normative emotional ratings for around 1000 English words. More recently VADER (Hutto & Gilbert, 2014) has emerged as a rule-based sentiment analysis tool that combines a lexicon and rule-based approach to measure sentiment.

In addition to these emotion-focused measures, stylometric features have traditionally included a range of measures like Readability and n-gram usage (Potthast et al., 2017) that represent different dimensions of linguistic style.

Recently Large Language Models (LLMs) have been increasingly utilized to represent linguistic style. Li et al. (2019) and Sousa et al. (2019) have demonstrated the effectiveness of LLMs like BERT, in modeling various aspects of linguistic style, including sentiment. However, while these LLM-based approaches have shown impressive results, they often lack interpretability. Additionally, there has been a limited focus on understanding sensorial style in these approaches.

### 2.2 SENSORIAL LINGUISTICS

Sensorial linguistics investigates the relationship between sensory perception and language, studying how different experiences and perceptions are represented using linguistic units (Winter, 2016). The field has traditionally focused on the five classical senses: visual, auditory, olfactory, gustatory, and

haptic. However, recent research has expanded this model to include interoception as a sixth sense (Lynott et al., 2020).

Winter et al. (2018) analyzed the distribution of sensorial language across different parts of speech (nouns, adjectives, and verbs) and found that visual language dominates across all categories. This aligns with Viberg (1983)'s proposed universal hierarchy of the senses, with vision at the top, followed by hearing, touch, smell, and taste.

Lynott et al. (2020) introduced the Sensorimotor Lexicon, a comprehensive resource containing sensory ratings for around 40,000 concepts across six sensory dimensions, including interoception.

### 2.3 SENSORIAL STYLE

While traditional stylometry has focused on analyzing patterns in language use through various computational and statistical techniques, it has largely overlooked sensorial language. Stylometric measures like LIWC and VADER are not specifically designed to capture patterns in sensorial language use.

Recently, methods have been proposed to analyze sensorial style. Kernot et al. (2016) proposed a method to analyze sensorial style by measuring the use of sensory adjectives. Khalid & Srinivasan (2022) introduced a method to measure sensorial style based on synaesthesia, or the propensity to replace one sensorial modality with another.

Prior works have focused on analyzing traditional stylometry and sensorial style independently, but there remains a gap in understanding how these two aspects of linguistic style interact. Our work aims to bridge this gap by proposing a novel approach that models the relationship between traditional linguistic style (as captured by LIWC features) and sensorial style.

## 3 METHODS

### 3.1 REPRESENTING SENSORIAL STYLE

Sensorial style is modeled and represented across a range of granularities. A synaesthesia-based approach has been used to model sensorial style at a high level (Khalid & Srinivasan, 2022). A high-level approach to modeling sensorial style focuses on patterns of sensory language-use across broader linguistic units or entire texts, rather than on individual words. In contrast, we model sensorial style at the word-level, which focuses on individual sensorial words and their relationships to other linguistic style features.

We represent a sensorial sentence as a one-hot encoding of the sensorial vocabulary. Khalid & Srinivasan (2022) have defined the sensorial vocabulary $V$ as a subset of 18,749 words from the Lancaster Sensorimotor Lexicon (Lynott et al., 2020). They consider a sentence to be sensorial if it has one or more sensorial words in it. We use this criterion and consider a sensorial sentence to have just one sensorial term. For example, 'it is a noisy room' has two sensorial words, the auditory 'noisy' and the visual 'room'. Assuming 'noisy' and 'room' are the second and fourth words in the sensorial vocabulary, this sentence constitutes two sensorial sentences represented as [0, 1, 0, 0, . . . , 0] for 'noisy' and [0, 0, 0, 1, . . . , 0] for 'room'. The length of the two vectors equals the size of our sensorial vocabulary; that is, $|V| = 18,749$.

We formalize the previous idea as follows. Let $V = \{w_1, w_2, ..., w_n\}$ be the sensorial vocabulary of size $n$. For a given sensorial word $w$ in a sentence, we represent it as a vector $\mathbf{y} \in \{0, 1\}^n$, where $y_i = 1$ if $w = w_i$ and 0 otherwise. A sentence $S$ with $m$ sensorial words is represented as a set of $m$ $n$-vectors and $S = \{\mathbf{y}_1, \mathbf{y}_2, ..., \mathbf{y}_m\}$, where $\mathbf{y}_j$ $(j = 1, \ldots, m)$ corresponds to the one hot encoding of the $j$th sensorial sentence.

We represent each sensorial sentence as a vector based on the LIWC-style (Pennebaker et al., 2015). Let $X = \{x_1, x_2, ..., x_m\}$ be the set of $m$ LIWC categories. For a given sensorial sentence $S$, we exclude the sensorial term $w_s$ and represent the style of the remaining sentence as a vector $\mathbf{s} \in \mathbb{R}^m$. Each element $s_i$ of this vector corresponds to the proportion of words in $S$ excluding $w_s$ that belong to the $i^{th}$ LIWC category $x_i$: $s_i = (|\{w \in S \setminus \{w_s\} : w \in x_i\}|)/(|S| - 1)$.

For example, given the sentence 'it is a noisy room' with two sensorial words 'noisy' and 'room', we create two style vectors. For 'room', the style vector will be based on ['it', 'is', 'a', 'noisy'], and for 'noisy' the style vector will be based on ['it', 'is', 'a', 'room']. This is comparable to the BERT masked language model setup, where each sensorial word is treated as the target word to be predicted, and the embedding is calculated from the remaining words in the sentence.

## 3.2 Linear Models for Style Interactions

We use regression to model the relation between traditional style and sensorial style. Let the style features of a sentence $S$ be the LIWC vector $\mathbf{x} = (x_1, \ldots x_m)$ and let $\mathbf{y} = (y_1, y_2 \ldots y_n)$ be the one-hot sensorial vector of the sentence, where $m$ is the number of style features and $n$ is the size of the sensorial vocabulary $\mathcal{S}$ (Lynott et al., 2020). Then, $\mathbf{y}^\top = \mathbf{x}^\top \mathbf{B} + \mathbf{e}^\top$ models the relation between linguistic style $\mathbf{x}$ and sensorial language use $\mathbf{y}$, with $\mathbf{e}$ denoting the errors independent of $\mathbf{x}$. The regression coefficient matrix is $\mathbf{B} \in \mathbb{R}^{m \times n}$, and its $(i, j)$the element $b_{ij}$ is the mean increase in the sensorial word $y_j$ for a unit increase in style feature $x_i$, given other features in $\mathbf{x}$ remain unchanged. The linear regression model is equivalent to a sensorial-word-prediction problem, where we predict the sensorial word $w_s$ in a sentence from the linguistic style of the remaining text. This method is analogous to the masked word prediction task used to train LLMs like BERT Devlin (2018).

We fit the regression model to the training data as follows. For a set of $k$ sentences, the $i$th sentence has sensorial vector $\mathbf{y}_i = (y_{i1}, \ldots, y_{in})$, and its corresponding style vector is $\mathbf{x}_i = (x_{i1}, \ldots, x_{im})$. The training data are represented as the $k \times n$ matrix $\mathbf{Y} = [\mathbf{y}_1, \ldots, \mathbf{y}_k]^\top$ and $k \times m$ matrix $\mathbf{X} = [\mathbf{x}_1, \ldots, \mathbf{x}_k]^\top$. For a sufficiently large $k$, the least squares estimate of $\mathbf{B}$ is $(\mathbf{X}^\top \mathbf{X})^{-1} \mathbf{X}^\top \mathbf{Y}$ (Qian et al., 2022). Previous works have shown that LIWC features have a low-rank structure (Geng et al., 2020). However, the standard least squares approach fails to capture this structure and the latent dependencies between the sensorial features and LIWC-style features, which correspond to the columns of $\mathbf{Y}$ and $\mathbf{X}$. This limitation is particularly significant because not all LIWC features capture the same amount of information. For example, the function category words are more informative than categories like fillers. The word categories have group behavior. For instance, in the LIWC features, first person singular is a subcategory of personal pronouns, whereas the ingestion category contains words like '*eat*' that also belong to the verb category.

## 3.3 Reduced-Rank Ridge Regression

We circumvent the previous limitations by assuming that $\mathbf{B}$ is a low-rank matrix. This assumption implies that the previous linear model becomes a reduced-rank regression model (Anderson, 1951), which assumes that $\mathbf{B}$ has a rank $r$ and $r \ll \min\{m, n\}$. In a sparse $\mathbf{B}$, a large fraction of the entries are 0, where $b_{ij} = 0$ denotes that $x_i$ and $y_j$ are not associated. Similarly, a row sparse $\mathbf{B}$ has $b_{ij} = 0$ for $j = 1, \ldots, n$ for many $i$s. If the $i$th row of $\mathbf{B}$ is zero, then $x_i$ is not associated with any sensorial word. To model a rank-$r$ $\mathbf{B}$, we set $\mathbf{B} = \mathbf{U}\mathbf{V}^\top$, where $\mathbf{U} = (u_1, u_2 \ldots u_r) \in \mathbb{R}^{m \times r}$ and $\mathbf{V} = (v_1, v_2 \ldots v_r) \in \mathbb{R}^{n \times r}$. By assuming row sparsity of $\mathbf{B}$, we can effectively select a subset of LIWC features that have the strongest associations with sensorial words across different contexts. This assumption is more appropriate for our goals of identifying the most influential LIWC features that contribute to sensorial language use.

Consider a reduced-rank model for regressing $\mathbf{Y}$ on $\mathbf{X}$. For a rank $r$, Chen & Huang (2012) propose a (row) sparse reduced-rank regression (SRRR) of $\mathbf{B}$ via $\mathbf{U}$ and $\mathbf{V}$ estimates as

$$\hat{\mathbf{U}}_s, \hat{\mathbf{V}}_s = \operatorname*{argmin}_{\mathbf{U} \in \mathbb{R}^{m \times r}, \mathbf{V}^\top \mathbf{V} = \mathbf{I}_r} \frac{1}{2}\|\mathbf{Y} - \mathbf{X}\mathbf{U}\mathbf{V}^\top\|_F^2 + \lambda \sum_{j=1}^{m} \|\mathbf{U}_j\|_2, \quad \hat{\mathbf{B}}_s = \hat{\mathbf{U}}_s \hat{\mathbf{V}}_s^\top, \qquad (1)$$

where $\hat{\mathbf{B}}_s$ is the SRRR estimate of $\mathbf{B}$, $\mathbf{I}_r$ is an $r \times r$ identity matrix, $\|\cdot\|_F$ is the Frobenius norm, and $\|\mathbf{U}_j\|_2$ is the group lasso penalty on the $j$th row of $\mathbf{U}$ (Yuan & Lin, 2006). Qian et al. (2022) develop an efficient algorithm for estimating $\mathbf{U}$ and $\mathbf{V}$ using the alternative minimization algorithm, which estimates $\mathbf{U}$ given $\mathbf{V}$ and vice versa. The group lasso norm on $\mathbf{U}$ rows implies that some of the $\mathbf{B}_s$ rows are zeros, but the estimation algorithm suffers from computational bottlenecks particularly when $k$ and $m$ are in the order of ten thousand.

We propose Reduced-Rank Ridge Regression (R4) as an efficient alternative to SRRR. The $\mathbf{B}$ matrix in our problem is not sparse because all stylistic features are associated with sensorial words, even

when their magnitudes are small; therefore, we replace the group lasso penalty on the $\mathbf{B}$ rows by a ridge penalty to obtain the R4 estimates of $\mathbf{U}$ and $\mathbf{V}$ as

$$\hat{\mathbf{U}}, \hat{\mathbf{V}} = \underset{\mathbf{U} \in \mathbb{R}^{m \times r}, \, \mathbf{V}^\top \mathbf{V} = \mathbf{I}_r}{\operatorname{argmin}} \frac{1}{2} \|\mathbf{Y} - \mathbf{X} \mathbf{U} \mathbf{V}^\top\|_F^2 + \lambda \sum_{j=1}^m \|\mathbf{U}_j\|_2^2, \quad \hat{\mathbf{B}} = \hat{\mathbf{U}} \hat{\mathbf{V}}^\top, \quad (2)$$

where $\hat{\mathbf{B}}$ is the R4 estimate of $\mathbf{B}$ and is obtained by a slight modification of the alternative minimization algorithm in Qian et al. (2022). The estimation algorithm of $\mathbf{V}$ given $\mathbf{U}$ remains the same in equation 1, but the estimation of $\mathbf{U}$ given $\mathbf{V}$ uses ridge regression. Unlike $\mathbf{B}_s$ in equation 1, $\hat{\mathbf{B}}$ is not sparse but has better predictive performance (Hastie, 2020). The columns of $\hat{\mathbf{U}}$ represent the latent factors or components that capture the shared structure between LIWC and sensorial features.

## 3.4 Modeling Non-Linear Style Interactions

The R4 model in equation 2 assumes a linear association between LIWC-style and sensorial style. The associations, however, are nonlinear from linguistic and cognitive perspectives. We model the relationship between LIWC-style and sensorial style as a phenomenon mediated by a *Central Language Processing Unit* (CLPU), using Large Language Models (LLMs) as a proxy for the of the *CLPU*. The *CLPU* is a similar construct to Levelt (1992)'s 'mental lexicon'. LLMs, trained on vast corpora of human language, encapsulate general language norms and patterns. They capture the complex interactions mediated by our broader linguistic knowledge and cognitive processes (Manning et al., 2020).

To model this interaction, we represent traditional stylistic features of a sentence using our LIWC-based representation. We then use an LLM for a masked language modeling task on the original sentence, with the sensorial words masked. Finally, we use the LLM's predictions for masked sensorial words, combined with the LIWC-style, to predict sensorial style. Formally, let $S$ be the original sentence, and $m(S)$ be the sentence with sensorial words masked. Let $f$ be the function represented by the LLM that takes the masked sentence $m(S)$ and returns the encoder embedding representation of the masked word. Then, the model relating sensorial words and LLM's encoder embeddings of the masked word is

$$\mathbf{y}_i = g(f(m(S_i)); \mathbf{x}_i) + \mathbf{e}_i, \quad \mathbf{e}_i \in \mathbb{R}^n, \quad i = 1, \ldots, k, \quad (3)$$

where $S_i$ is the $i$th sentence, $\mathbf{y}_i$ and $\mathbf{x}_i$ remain the same as in equation 2, $\mathbf{e}_i$ is the $i$th error vector, and $g$ is a classifier function that predicts sensorial language use from the combination of the LLM's encoder embeddings and the original stylistic features.

## 3.5 Stylometrically Lean Interpretable Models (SLIM-LLMs)

LLMs like BERT are often overparameterized (Matton & de Oliveira, 2019). This can obscure the relationship between LIWC-style and sensorial style due to redundancies in the model's training. To address this, we propose using dimensionality reduction techniques to create Stylometrically Lean Interpretable Models (SLIM-LLMs). SLIM-LLMs are reduced versions of standard LLMs that aim to reveal the underlying relationships between LIWC-style and sensorial style more clearly. We create SLIM-LLMs using Singular Value Decomposition (SVD). Let $\mathbf{E} \in \mathbb{R}^{k \times d}$ be the encoder embedding matrix of our LLM, where $d$ is the dimension of the hidden state and $k$ is the number of sentences in our dataset.

The SLIM-LLM retain only the top $r$ singular values and their corresponding singular vectors for the SVD of $\mathbf{E}$ and are denoted as $\mathbf{E}_{\text{slim}}$. Specifically, let $\mathbf{E} = \mathbf{U}\boldsymbol{\Sigma}\mathbf{V}^\top$ be the SVD of $\mathbf{E}$, where $\mathbf{U} \in \mathbb{R}^{k \times k}$ and $\mathbf{V} \in \mathbb{R}^{d \times d}$ are the left and right orthonormal matrices. Then, $\mathbf{E}_{\text{slim}} = \mathbf{U}_r \boldsymbol{\Sigma}_r \mathbf{V}_r^\top$, where $\mathbf{U}_r \in \mathbb{R}^{k \times r}$, $\boldsymbol{\Sigma}_r \in \mathbb{R}^{r \times r}$, and $\mathbf{V}_r \in \mathbb{R}^{d \times r}$. The nonlinear classification model relating sensorial words and LLMs in equation 3 is now rewritten for SLIM-LLMs as

$$\mathbf{y}_i = g(f_{\text{slim}}(m(S_i)); \mathbf{x}_i) + \mathbf{e}_{\text{slim}i}, \quad \mathbf{e}_{\text{slim}i} \in \mathbb{R}^n, \quad i = 1, \ldots, k, \quad (4)$$

where $\mathbf{e}_{\text{slim}i}$ is the $i$th error term, $f_{\text{slim}}$ is the function represented by our SLIM-LLM that takes the masked sentence $m(S_i)$ as input and outputs a dimension-reduced embedding of $\mathbf{x}_i$, and $g$ is a classifier function that predicts sensorial language use from the combination of the SLIM-LLM's reduced encoder embeddings and the original stylistic features. In this formulation, $f_{\text{slim}}(m(S_i))$ represents

the projection of the masked sentence $m(S_i)$ onto the reduced-dimensional space defined by $\mathbf{U}_r$ so that $f_{\text{slim}}(m(S_i)) = \mathbf{U}_r^\top f(m(S_i))$, where $f(m(S_i))$ is the original LLM's encoder embedding for the masked sentence $m(S_i)$. By reducing the dimensionality of the encoder embeddings, we aim to maintain the benefits of using LLMs as proxies for the mental lexicon while revealing more interpretable relationships between the different aspects of linguistic style.

The choice of $r$, the number of singular values to retain, represents a trade-off between model complexity and interpretability. A smaller $r$ results in a more interpretable model, but may lose some nuanced relationships, while a larger $r$ retains more information but may be less interpretable. The optimal value of $r$ can be determined through empirical analysis.

## 4 EXPERIMENTS

We use BERT-base (Devlin, 2018) to investigate the relationship between traditional style (LIWC-style) and sensorial style across diverse contexts. We study the style of 5 different text genres[1]. This section details the datasets and models used in our study.

### 4.1 DATASETS

| Language Genre | Datasets | Source | Sensorial Sentences |
|---|---|---|---|
| Critical | Business Reviews | Yelp.com | 2,101,603 |
| Literary | Novels | Project Gutenberg | 1,929,260 |
| Poetic | Music Lyrics | Genius.com | 1,107,749 |
| Persuasive | Advertisements | Airbnb Descriptions | 1,442,050 |
| Informative | Articles | Wikipedia | 1,563,888 |

Table 1: Overview of text collections and genres

We analyze 5 different text genres. Each language genre represents a distinct way in which language is employed to achieve specific communicative goals or to serve particular purposes.

**Critical Language:** Reviews from the Yelp Dataset Challenge (2005-2013), encompassing approximately 42,000 businesses.

**Literary Language:** English novels from Project Gutenberg's Domestic fiction category, spanning works from $18^{th}$ century author Regina Maria Roche to $20^{th}$ century writer Lucy Maud Montgomery.

**Poetic Language:** Lyrics of songs featured on the Billboard Hot 100 charts (1963-2021), obtained via the Genius API. This chart is widely regarded as the music industry benchmark (Whitburn, 2010).

**Persuasive Language:** Airbnb property descriptions (2008-2022), showcasing accommodations, amenities, and local attractions to potential guests.

**Informative Language:** Wikipedia articles, collected in July 2024. Unlike other datasets, these entries are subject to continuous updates, precluding precise dating.

Table 1 presents an overview of our text collections and genres, along with the specific number of sensorial sentences extracted from each collection. For our experiments, we randomly select a standardized sample of 300,000 sensorial sentences from each set to ensure consistency across all language aspects.

---

[1]Experiments using BERT-large, DistilBERT (Sanh et al., 2019) and RoBERTa-base (Liu, 2019) gave comparable results (See: Appendix A.2), thus we only report BERT-base results.

## 4.2 RESULTS

### 4.2.1 LATENT REPRESENTATION OF LIWC-STYLE

We investigate the relationship between the latent representation of LIWC-style and sensorial style. To find the optimal number of latent dimensions that best capture LIWC-style, we solve the Reduced-Rank Ridge Regression (R4) for a range of $r$ values from 1 to 74.

Using the reconstructed $\mathbf{B} = \mathbf{U}\mathbf{V}^\top$ for this range of $r$, we calculate the mean squared error (MSE) on the test data. Figure 1 shows the MSE for the five datasets across different values of $r$.

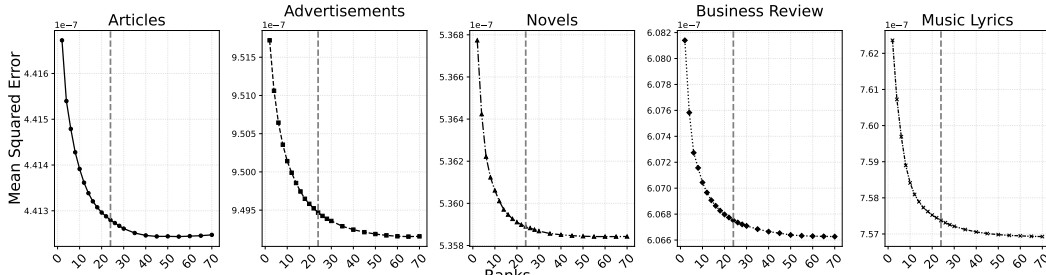

Figure 1: Mean Squared Error (MSE) for the five language aspect datasets (Articles, Advertisements, Novels, Business Reviews, and Music Lyrics) plotted against the number of latent dimensions ($r$) in the Reduced-Rank Ridge Regression (R4) model. The plot shows the decrease in reconstruction error as the number of latent dimensions increases from 1 to 74.

While the reconstruction errors vary in absolute terms between the five genres, we observe a general trend across all datasets. On average, we see the greatest decrease in the reconstruction error within the first 20 dimensions. The error rate begins to asymptote for values of $r > 20$.

Based on this observation and the diminishing returns in error reduction, we empirically determine that $r \approx 24$ provides an optimal latent dimension representation for LIWC-style. This choice balances model complexity with performance, capturing most of the variance in the data while maintaining interpretability.

This finding suggests that the relationship between LIWC-style features and sensorial language use can be effectively represented in a relatively low-dimensional latent space across diverse language genres.

### 4.3 GROUP STRUCTURE IN LIWC-STYLE

In the original formulation of our model, $\mathbf{y}^\top = \mathbf{x}^\top \mathbf{B} + e^\top$, all dimensions of the LIWC features are treated as independent. However, our analysis of the $\mathbf{U} \in \mathbb{R}^{n \times r}$ matrix, which represents the latent dimensions of our Reduced-Rank Ridge Regression (R4) model, reveals group structures indicating inter-dependencies among LIWC features and their collective relationship with sensorial style.

Figure 2 illustrates the group structure in the $\mathbf{U} \in \mathbb{R}^{74 \times 24}$ latent representation for Wikipedia articles[2]. We find similar group structures in the latent representations of other genres as well. From the figure, we note that

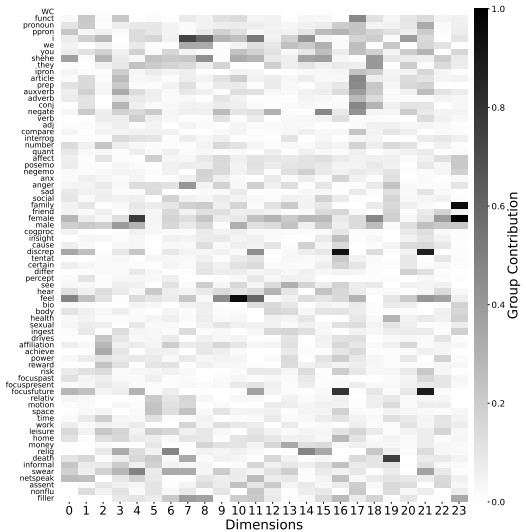

Figure 2: Heatmap showing the latent representation of LIWC categories across 24 dimensions for Wikipedia articles. The intensity indicates the strength of the contribution of each LIWC category to each latent dimension.

---

[2]See Appendix A.1 for the representations of other genres in more detail visualizations.

some latent dimensions appear more influential than others, as indicated by stronger and more widespread contributions across LIWC categories, as an example the Discrepancy category *'discrep'* contributes to both groups 16 and 21. We also find that related LIWC categories often contribute strongly to the same latent dimensions, forming natural groupings. An example of this would be the contribution of function words, categories like *'i'*[3], *'we'*[4], *'shehe'*[5] in Group 17.

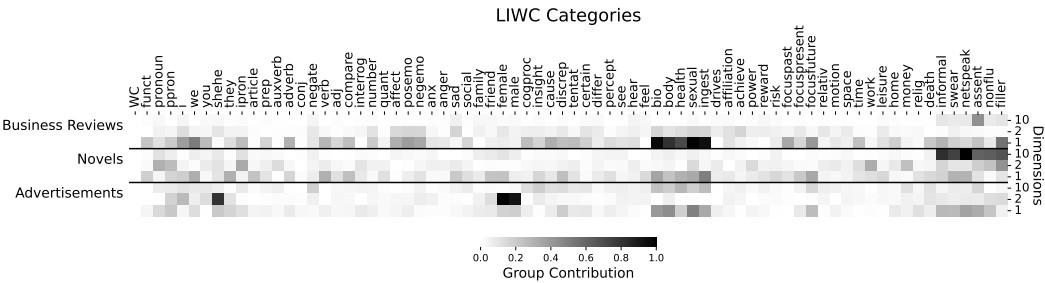

Figure 3: The heatmap shows the contribution of LIWC categories to specific latent dimensions, across three genres: Business Reviews, Novels, and Advertisements.

In Figure 3, we examine a sample of columns of 3 other genres. We observe that:

**Business Reviews (Yelp):** A group forms around categories of LIWC biological processes, including words focused on consumption. This aligns with the nature of restaurant reviews, where descriptions of food and eating experiences are central.

**Novels (Gutenberg):** We observe a group forming around informal language use, including categories related to fillers, non-fluencies, and netspeak. This clustering would reflect the author's attempt to mimic natural, conversational speech patterns in dialogue and narration.

**Advertisements (Airbnb):** We observe an emergent group that combines elements from disparate LIWC categories, specifically gendered words (masculine and feminine) from the social processes category and gendered pronouns (she/he) from the function word category. This grouping is not apparent in the standard LIWC classification but emerges in our analysis. Such a pattern suggests that Airbnb property descriptions may employ gender-specific language strategies that are not captured by LIWC's predefined categories. This finding demonstrates how our approach can reveal latent linguistic structures that are not immediately evident from simple LIWC groupings, potentially offering new insights into the stylistic techniques used in persuasive advertising language.

These groupings, emerging from the latent representation, reveal how different aspects of language use cluster together in genre-specific ways. They provide insights into the underlying structures of LIWC-style across various text types and how these relate to sensorial style. The presence of these group structures, not accounted for in the original independent dimension assumption, highlights the complexity of the relationship between LIWC-style features and sensorial style.

## 4.4 EXPLORING LIWC-STYLE USING SLIM-BERT

We investigate the relationship between linguistic style and sensorial language use by using low-dimensional projections of LLMs — SLIM-LLMs model augmented with LIWC features. We use these SLIM-LLMs for the sensorial word prediction task described in section 3.4. For each masked sensorial sentence, we extract the SLIM-LLM representation and use it (along with LIWC representations) as input to a fully connected Multi-Layered Perceptron (MLP) that is trained to predict the

---

[3] 1st person pronouns.

[4] 3rd person pronouns.

[5] 2nd person pronouns.

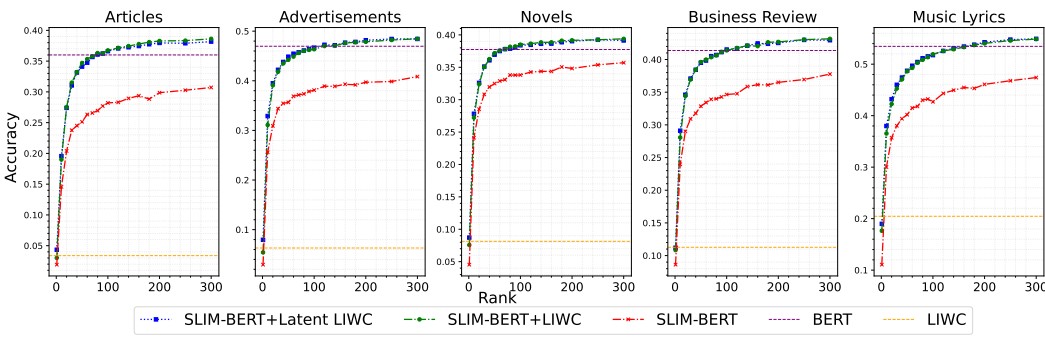

Figure 4: Accuracy of sensorial word prediction against the rank (number of dimensions) used in the SLIM-BERT model for different language aspects

masked sensorial word. Figure 4 presents the performance of BERT-base for each language aspect. We focus on the first 240 dimensions of the SLIM-BERT model.

We compare the performance of three configurations of SLIM-BERT:

**SLIM-BERT+Latent LIWC**: SLIM-BERT augmented with latent LIWC features.

**SLIM-BERT+LIWC**: SLIM-BERT augmented with raw LIWC features.

**SLIM-BERT**: SLIM-BERT without LIWC features.

For reference, we also show the performance of the full BERT-base model and raw LIWC features (shown as horizontal lines).

Across all genres, we observe that augmenting SLIM-BERT with LIWC features (both latent and raw) consistently improves performance over SLIM-BERT alone. For instance, in Articles, SLIM-BERT+Latent LIWC achieves an accuracy of 0.380, compared to 0.299 for SLIM-BERT alone. This pattern is consistent across other categories, with SLIM-BERT+Latent LIWC reaching accuracies of 0.483 for Advertisements, 0.390 for Novels, 0.430 for Business Reviews, and 0.545 for Music Lyrics. These results suggest that linguistic style, as captured by LIWC, provides complementary information to the language model for predicting sensorial language use.

The SLIM-BERT with the Latent LIWC configuration performs as well as or slightly better than SLIM-BERT with the raw LIWC features. For example, in the Music Lyrics category, SLIM-BERT+Latent LIWC achieves 0.545 accuracy compared to 0.543 for SLIM-BERT+LIWC, indicating that the latent representation of LIWC features effectively captures the most relevant aspects of linguistic style for this task, while potentially reducing noise or redundancy in the raw LIWC features.

In most cases, our SLIM-BERT+Latent LIWC configuration approaches or even exceeds the performance of the full BERT model, while using a fraction of the parameters. For instance, in Novels, SLIM-BERT+Latent LIWC achieves 0.390 accuracy compared to 0.378 for the full BERT model. Similarly, for Business Reviews, SLIM-BERT+Latent LIWC reaches 0.430 accuracy, surpassing the full BERT model's 0.416. This demonstrates the effectiveness of our dimensionality reduction approach in capturing the most relevant features for this task. The dimensionality reduction filters out noise and less relevant information, focusing on the most salient features of sensorial language prediction. Additionally, the addition of latent LIWC features provides complementary stylistic information that enhances our model's predictive power.

These results demonstrate the effectiveness of our SLIM-BERT approach in modeling the relationship between linguistic style and sensorial language use. The consistent improvements from LIWC augmentation, particularly using our latent LIWC representation, suggest a strong link between stylometric features and sensorial language across various language aspects. This supports our hypothesis of a mediated interaction between linguistic style and sensorial language, as modeled by our SLIM-LLM framework.

## 5 DISCUSSION

Our results demonstrate that SLIM-LLMs, such as SLIM-BERT, and LIWC-style features capture complementary aspects of sensorial style across various language genres. The combination of these two representations consistently outperforms either representation alone, supporting our hypothesis of a mediated interaction between LIWC-style and sensorial style through a *Central Language Processing Unit* (CLPU).

For example, in the case of Articles and Advertisements, we observe that the combination of SLIM-BERT and Latent LIWC features achieves higher accuracy than the sum of their individual performances. Specifically, for Articles, SLIM-BERT+Latent LIWC with $r = 240$ achieves an accuracy of 0.48, compared to SLIM-BERT (0.41 at $r = 240$) and LIWC-style (0.06) alone.

While we focused on LIWC-style features in this work, our approach can be extended to incorporate other stylometric features such as ANEW, VADER, and measures of linguistic complexity like Readability and Hapax Legomenon. Such extensions would let us not only study the relationships between these features and sensorial style, but also the interactions with the rest of the stylometric features.

One limitation of this study is its focus on English language texts. However, the dimensionality reduction technique used to create SLIM-LLMs is not inherently language-specific and is only limited by the underlying LLM's training data. This approach can be extended to other languages by creating SLIM versions of language-specific or multilingual models, such as SLIM-BETO for Spanish (based on the BETO model (Cañete et al., 2020)) or SLIM-mBERT (based on the multilingual BERT model).

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

# A APPENDIX

## A.1 LATENT REPRESENTATIONS OF LIWC-STYLE ACROSS TEXT GENRES

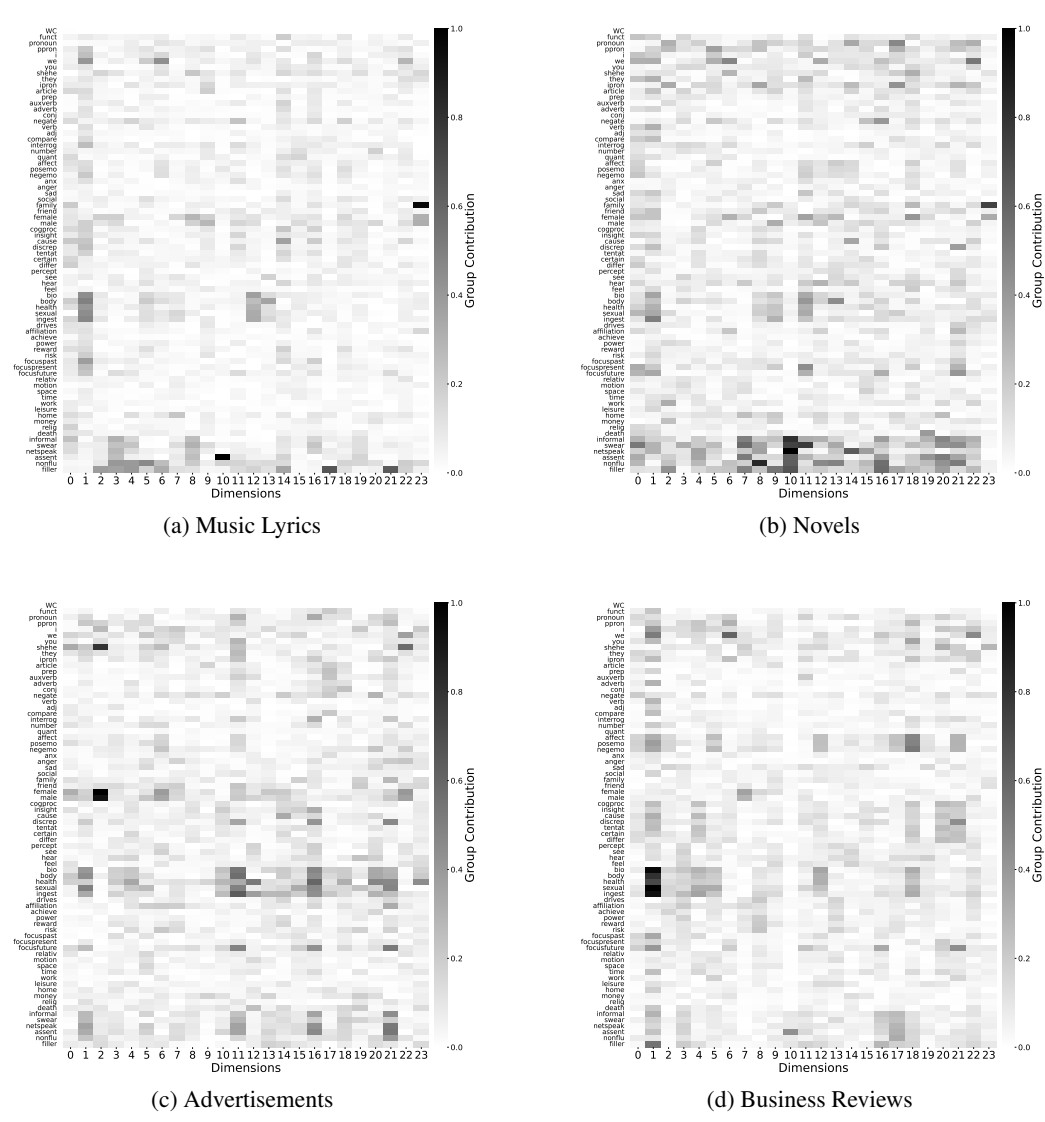

(a) Music Lyrics  (b) Novels

(c) Advertisements  (d) Business Reviews

Figure 5: Heatmaps showing the latent representation of LIWC categories across 24 dimensions for different text genres: (a) Music Lyrics, (b) Novels, (c) Advertisements, and (d) Business Reviews. The intensity indicates the strength of contribution of each LIWC category to each latent dimension.

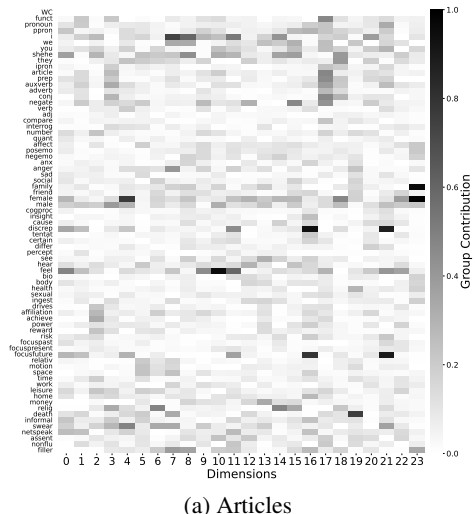

(a) Articles

Figure 6: Heatmap showing the latent representation of LIWC categories across 24 dimensions for (e) Articles.

## A.2 PERFORMANCE COMPARISON OF SLIM-LLMS ACROSS DIFFERENT TEXT GENRES

| Model | Configuration | Articles | | | Advertisements | | | Novels | | | Business Reviews | | | Music Lyrics | | |
|---|---|---|---|---|---|---|---|---|---|---|---|---|---|---|---|---|
| | | SLIM-80 | SLIM-240 | Full | SLIM-80 | SLIM-240 | Full | SLIM-80 | SLIM-240 | Full | SLIM-80 | SLIM-240 | Full | SLIM-80 | SLIM-240 | Full |
| BERT-base | Full Model | – | – | 0.360 | – | – | 0.469 | – | – | 0.378 | – | – | 0.416 | – | – | 0.533 |
| | SLIM | 0.279 | 0.299 | – | 0.372 | 0.403 | – | 0.332 | 0.352 | – | 0.339 | 0.368 | – | 0.423 | 0.465 | – |
| | SLIM + LIWC | 0.362 | **0.385** | – | 0.457 | 0.481 | – | 0.381 | **0.391** | – | 0.407 | 0.429 | – | 0.511 | 0.543 | – |
| | SLIM + Latent LIWC | 0.357 | 0.380 | – | 0.462 | **0.483** | – | 0.379 | 0.390 | – | 0.409 | **0.430** | – | 0.510 | **0.545** | – |
| BERT-large | Full Model | – | – | 0.373 | – | – | 0.473 | – | – | 0.389 | – | – | 0.420 | – | – | 0.517 |
| | SLIM | 0.289 | 0.315 | – | 0.380 | 0.418 | – | 0.348 | 0.364 | – | 0.355 | 0.380 | – | 0.434 | 0.464 | – |
| | SLIM + LIWC | 0.384 | **0.404** | – | 0.469 | **0.491** | – | 0.394 | **0.406** | – | 0.424 | **0.440** | – | 0.513 | 0.540 | – |
| | SLIM + Latent LIWC | 0.379 | 0.400 | – | 0.473 | **0.491** | – | 0.393 | 0.405 | – | 0.424 | 0.439 | – | 0.514 | **0.542** | – |
| RoBERTa | Full Model | – | – | 0.356 | – | – | 0.499 | – | – | 0.397 | – | – | 0.465 | – | – | 0.565 |
| | SLIM | 0.242 | 0.281 | – | 0.386 | 0.418 | – | 0.327 | 0.356 | – | 0.367 | 0.401 | – | 0.438 | 0.489 | – |
| | SLIM + LIWC | 0.336 | **0.365** | – | 0.472 | 0.501 | – | 0.386 | **0.405** | – | 0.440 | **0.467** | – | 0.525 | 0.566 | – |
| | SLIM + Latent LIWC | 0.336 | 0.363 | – | 0.478 | **0.502** | – | 0.387 | 0.403 | – | 0.441 | 0.466 | – | 0.528 | **0.567** | – |
| DistilBERT | Full Model | – | – | 0.330 | – | – | 0.454 | – | – | 0.348 | – | – | 0.391 | – | – | 0.523 |
| | SLIM | 0.237 | 0.267 | – | 0.345 | 0.378 | – | 0.290 | 0.318 | – | 0.305 | 0.332 | – | 0.397 | 0.446 | – |
| | SLIM + LIWC | 0.326 | 0.347 | – | 0.442 | **0.467** | – | 0.343 | 0.359 | – | 0.376 | **0.407** | – | 0.493 | 0.532 | – |
| | SLIM + Latent LIWC | 0.327 | **0.350** | – | 0.441 | **0.467** | – | 0.344 | **0.361** | – | 0.378 | 0.400 | – | 0.494 | **0.533** | – |
| LIWC | – | – | – | 0.033 | – | – | 0.063 | – | – | 0.083 | – | – | 0.113 | – | – | 0.203 |

Note: SLIM-80 and SLIM-240 refer to SLIM-LLM models with 80 and 240 dimensions respectively. The best performing SLIM-LLM configuration for each model and genre is highlighted in bold.

