# OpenReview forum: "SLIM-LLMs: Low-Rank Models of Linguistic Style"
_ICLR.cc/2025/Conference — Submitted to ICLR 2025_

### Official Review · Reviewer_jB1E · 2024-11-01

**Soundness:** 2
**Presentation:** 2
**Contribution:** 2
**Rating:** 3
**Confidence:** 4

**Summary:**

The paper introduces a new approach to model sensorial stylometry. Drawing inspiiration from cognition theory, the paper models sensory stylometry as a combination of traditional stylometry dimensions. Further, this is modelled using a Reduced Rank Ridge Regression formulation - where each "sensorial" sentence is modelled as a low-ranked weighted combination of the tradtional style dimension. To extend to non-linear mappings, masked LLM representation is used instead of the direct encoding - allowing for complex relations to be captured. Experiments indicate that this is better than directly using BERT representations, while having lesser parameters.

**Strengths:**

The paper has an interesting formulation of style and proposes a solution with a strong statistical foundation for the problem.

**Weaknesses:**

While the paper has merits, the paper should be improved along several dimensions before being ready for acceptance:
1) The notion of sensory stylometry is never established. Given style is a qualitative element, I would have liked to see a few examples articulating this clearly. For example, the introduction of the paper can include a few examples that establish what is sensory stylometry - this will help establish the problem better.
2) The paper grounds its observations on a lot of quantitative evaluation, which is severely limiting given the qualitative nature of the task. For example, it would be good to provide a few examples where the linear model would be sufficient, and examples where non-linear model will be needed. This will help in articulating the strengths and intuitions of the proposed framework.

**Questions:**

See above

---

### Official Review · Reviewer_Nt9H · 2024-11-04

**Soundness:** 3
**Presentation:** 2
**Contribution:** 2
**Rating:** 5
**Confidence:** 3

**Summary:**

The paper starts with a review of the literature on style features.  Then in section 3, the paper switches gears and discusses reduced-rank ridge regression.  It wasn't clear to this reviewer until around section 4 where it was going.  The datasets in Table 1 look more like genres than style.  In fact, the paper uses that word there.  The results in Figure 4 show that the proposed method is better than alternatives at predicting the next word in 5 genres.  These results demonstrate the effectiveness of our SLIM-BERT approach in modeling the relationship between linguistic style and sensorial language use.

**Strengths:**

Corpora such as the Brown Corpus and the British National Corpus took genre very seriously.  They used to talk about balanced corpora.  It would be nice to see those considerations taken more seriously.

The proposed method looks very reasonable.

**Weaknesses:**

The presentation is unnecessarily hard going.  It was not clear where the argument was going.  The discussion of style in the beginning suggested to this reviewer that the paper may have been sent to the wrong venue.  It was only later that I saw some machine learning.

Even so, genre and style are very different things.  It isn't clear to me that the proposed method is really capturing either.  I would feel happier with more modest claims suggesting that it is worth taking genre more seriously than many LLMs do.  I think you have a convincing case for the more modest conclusions.

It is a shame that Figure 2 falls on top of footnote 2.  The page break doesn't work well with the wrapped figure.

Figure 2 is hard to read.  Please don't use less than 8pt fonts.  I'm not sure what I am supposed to take-away from figure 2 and figure 3.  I get that different genres are different.  Is there more to be said than that.  Would it help to embed the genres in latent dimensions and show scatter plots of the first few latent dimensions?   As I recall, Biber did quite a bit of work like that.  It would be good to reformulate that way of thinking using more modern methods.

**Questions:**

Would it be worth while citing work on balanced corpora such as the Brown Corpus and the British National Corpus?

Ditto for Biber's work.

Could you provide a road map early on with an overview of where the paper is going?

Can you explain what you mean by style and genre?  I was under the impression that they are very different, but maybe you have a different definition in mind.

---

> ### Comment · Reviewer_Nt9H · 2024-11-26
> **not much to add**
>
> I think we have discussed this sufficiently.  The system is complaining that I am not commenting enough so I am adding this stupid non-comment.
>
> Biber (https://scholar.google.com/citations?user=mdWIU4MAAAAJ&hl=en) has an h-index of 94.  We should be aware of that work.  He talks a lot about style and genre and more.
>
> The figure is obviously unreadable.  That is obvious.  My point was to ask the authors to address that.  I don't think more needs to be said about that.  Please do not use less than 8 point font anywhere.
>
> A google search on the the British National Corpus and the Brown Corpus will return plenty of highly cited papers.  The wikipedia page discusses balance in some detail: https://en.wikipedia.org/wiki/Brown_Corpus.  I would have thought that the hints in my review would have been sufficient to find this literature.
>
> I checked the box for first time reviewer because this may be the first time that I have reviewed for this venue, but I have reviewed quite a bit.

---

### Official Review · Reviewer_HYBt · 2024-11-04

**Soundness:** 3
**Presentation:** 3
**Contribution:** 2
**Rating:** 3
**Confidence:** 5

**Summary:**

This paper introduces SLIM-LLMs (Stylometrically Lean Interpretable Models), a novel approach to modeling relationships between traditional linguistic style and sensorial language use. The key innovation is using Reduced-Rank Ridge Regression (R4) to show that these relationships can be effectively captured in just 24 dimensions, rather than the original 74 LIWC features. The authors demonstrate that SLIM-LLMs can achieve comparable performance to full-scale language models while using up to 80% fewer parameters, validated across five different text genres.

**Strengths:**

The authors propose a framework that bridges traditional stylometrics and sensorial linguistics, providing a cognitively-motivated framework for understanding style interactions. The theoretical framework is the main contribution of the paper.

Introduces Reduced-Rank Ridge Regression (R4) as an efficient alternative to SRRR

Develops SLIM-LLMs that achieve comparable performance with significantly fewer parameters

The analysis of features for different domains is novel and of interest

**Weaknesses:**

The computational models introduced in this paper are of limited novelty, as a result the overall contribution of this paper is limited: the breakdown of stylometric vs sensorial while theoretically interesting is not clear how it improves computationally the model; the proposed R4 method is a slight variation on existing methods.

Limited exploration of alternative dimension reduction techniques: The fact that dimensionality reduction on LIWC works is not surprising. It would be great to compare with other methods of dimensionality reduction and show that r=24 for R4 is the best compression you can achieve.

Show computationally that the stylometric vs sensorial has an impact on your model.

No human evaluation of the interpretability claims

Only considers one-hot encoding for sensorial words

**Questions:**

Can you provide additional evidence on the computational importance of breaking down stylometric vs sensorial features?

Why is R4 the best dimensionality reduction technique for LIWC features for this task? How about for other tasks, e.g., polarity detection?

---

### Official Review · Reviewer_zKt4 · 2024-11-04

**Soundness:** 2
**Presentation:** 3
**Contribution:** 3
**Rating:** 5
**Confidence:** 3

**Summary:**

This paper studies the relationship between the sensorial styles and the other stylistic features in LIWC by modeling their interactions with linear regression models and BERT. It also proposes SLIM-LLM that uses low-dimensional latent representations to effectively capture the stylistic information pertinent to sensorial language prediction across five text genres.

**Strengths:**

This work studies the relationship between sensorial language and traditional stylometrics, which has the potential to provide insights into textual analysis. The authors may further highlight their contributions by discussing the potential application (e.g., stylometric studies, writing analysis, settings where expressive language is examined) of their study.

**Weaknesses:**

* The phrasing like "the relationship between traditional style (LIWC-style) and sensorial style" may not be accurate. They are not totally independent, and LIWC-style includes categories that can capture aspects of sensory style.
* Apart from applying SVD to the BERT embedding, have the authors considered freezing some layers of the model while only training a few layers? Or other parameter-efficient methods such as LoRA? These methods are natural to think about and could provide a valuable basis for experimental comparison.
* This paper lacks the recognition of other works to provide a better academic background of the study. There has been some relevant work on using sensorial style together with other dimension of linguistic style for text analysis (https://doi.org/10.1080/09296174.2017.1405719, https://dl.acm.org/doi/pdf/10.1145/1979742.1979614), and there are also other recent works on improving LLMs' understanding of diverse linguistic style using lexicon (https://aclanthology.org/2024.acl-long.740/). It would be beneficial to cite and discuss these works to highlight the differences from them and contextualize the findings.
* The structure of the sections, particularly Section 4, appears disorganized and rushed. It may be beneficial for the author to consider reorganizing them for better clarity.
* The analysis in Section 4 seems superficial. It mainly focuses on the grouping of LIWC features and how these groups manifest in different text genres. However, it does not show how these categories might affect the personal engagement with sensorial description.

**Questions:**

* In Section 4.1 (table 1), how were the sensorial sentences extracted from each of the original datasets? How were sensorial words annotated?
* The footnote and Figure 2 caption in page 7 are vague.
* What are training details of Figure 4? For example, what is the learning rate of BERT? Did the authors explore hyperparameter tuning? Given the concern that BERT may be overparameterized (as also mentioned in paper), adjusting hyperparameters could significantly affect its performance.

---

### Meta-Review · Area_Chair_HsqH · 2024-12-20

**Metareview:**

The paper proposes Stylometrically Lean Interpretable Models (SLIM-LLMs) which use dimensionality-reduced LLMs with LIWC features reduced to 24 dimensions instead of the original 74. The main insight from the investigation is showing the effect of  relationship between linguistic style and sensorial language using SLIM-LLMs model augmented with LIWC features. This relation is studied by linear regression and is tested using three configuration of SLIM-BERT with and without LIWC features.

**Additional Comments On Reviewer Discussion:**

The reviewer pointed out numerous issues with the paper, from unclear presentation, to limited novelty, missing evaluations and more. The authors did not provide response to the reviewer comments. The ranking from all reviewers was below acceptance threshold.

---

### Decision · Program_Chairs · 2025-01-22

Reject